# A Concise Review of Liquid Chromatography-Mass Spectrometry-Based Quantification Methods for Short Chain Fatty Acids as Endogenous Biomarkers

**DOI:** 10.3390/ijms232113486

**Published:** 2022-11-03

**Authors:** Neerja Trivedi, Helen E. Erickson, Veenu Bala, Yashpal S. Chhonker, Daryl J. Murry

**Affiliations:** 1Clinical Pharmacological Laboratory, Department of Pharmacy Practice and Science, University of Nebraska Medical Center, Omaha, NE 68198, USA; 2Department of Pharmaceutical Sciences, University of Nebraska Medical Center, Omaha, NE 68198, USA; 3Fred and Pamela Buffett Cancer Center, University of Nebraska Medical Center, Omaha, NE 68198, USA

**Keywords:** short chain fatty acids, biological samples, biomarkers, LC-MS/MS

## Abstract

Fatty acids are widespread naturally occurring compounds, and essential constituents for living organisms. Short chain fatty acids (SCFAs) appeared as physiologically relevant metabolites for their involvement with gut microbiota, immunology, obesity, and other pathophysiological functions. This has raised the demand for reliable analytical detection methods in a variety of biological matrices. Here, we describe an updated overview of sample pretreatment techniques and liquid chromatography-mass spectrometry (LC-MS)-based methods for quantitative analysis of SCFAs in blood, plasma, serum, urine, feces and bacterial cultures. The present review incorporates various procedures and their applications to help researchers in choosing crucial parameters, such as pretreatment for complex biological matrices, and variables for chromatographic separation and detection, to establish a simple, sensitive, and robust quantitative method to advance our understanding of the role of SCFAs in human health and disease as potential biomarkers.

## 1. Introduction

Fatty acids belong to a class of lipids composed of carboxylic acids containing linear or branched aliphatic chains of varying length. They are classified based on carbon content: long chain fatty acids (C > 10), medium chain fatty acids (C5–C10), and short chain fatty acids (C2–C4) [1]. Fatty acids serve as building blocks for numerous biological constituents such as the phospholipids that form cellular membranes, energy storing long- and medium-chain triglycerides, and sphingolipids, which assume various roles in cell signaling [2,3]. Short chain fatty acids (SCFAs) are distinct from long and medium chain FAs because they are produced predominantly from intestinal microbial fermentation of fibers and resistant starches and absorbed with the aid of transport proteins highly expressed along the gastrointestinal tract, lymphocytes, kidney and thyroid [4,5,6,7]. SCFAs have been shown to engage with G-protein coupled receptor (GPCR) signaling pathways and inhibit histone deacetylases (HDACs) involved in regulating immune response, metabolism and energy storage [8,9,10,11,12,13]. As there is a growing body of evidence emphasizing the link between the human gut microbiome and chronic metabolic and inflammatory diseases, it is clear the presence of SCFAs in humans has significant implications for overall health [14,15,16,17,18].

SCFAs have diverse roles in physiological processes that promote human gut, musculoskeletal, and even brain health [19]. Acetic acid (AA), butyric acid (BA) and propionic acid (PA)–commonly referred to as their salt forms: acetate, butyrate, and propionate—are the main SCFAs produced by gut bacteria [20]. Acetate, an important pH regulator, accounts for the highest percentage of gut SCFAs [7,20]. Propionate has cholesterol-lowering, anti-cancer and anti-inflammatory properties, and contributes to satiation and regulation of blood pressure [21,22,23]. The least abundant of the three, butyrate, is the major energy source for the gut-lining colonocytes, and a potent HDAC inhibitor [24]. Butyrate has been found to induce differentiation of colonic regulatory T cells in mice, combat inflammation responsible for chronic diseases in humans, induce expression of detoxifying protein glutathione-*S*-transferase (GST), and prevent colon cancer by stimulating apoptosis of rogue cells [25,26,27,28,29]. Butyrate and other SCFAs are postulated to have neuroprotective effects via gut-brain axis signaling [30,31]. 

A clinical understanding of the relationship between the gut microbiota and disease progression relies on our ability to measure microbial byproducts. There is a need to establish the pharmacokinetics (PK) and pharmacodynamics (PD) profiles of SCFAs in humans, urging the importance of reliable quantitative methods. Biomarkers studies of SCFAs in several biological matrices (blood, plasma, serum, urine, and feces) have utilized analytical techniques such as gas chromatography (GC) and reverse phase (RP) liquid chromatography coupled to mass spectrometry (LC-MS) [1,32,33,34,35,36,37,38]. GC-MS is a robust method to detect SCFAs, but sample preparation is time-consuming, and its applicability is limited to a small number of biological samples [1,39,40]. The LC-MS approach requires less time for sample preparation and analysis. For example, the analysis time of ultra-high-pressure LC (UHPLC) is approximately one-fourth of GC without compromising resolution [32]. LC can be run at lower temperatures, limiting degradation of volatile SCFAs. In the past decade, innovations in sample extraction and chemical derivation have placed LC-MS analysis as the most sensitive and efficient method for metabolite detection [36,41]. To the best of our knowledge, this is the first review of LC-MS-based quantitation of SCFAs in assorted biological matrices, including plasma, serum, feces, urine, and cell culture. We present a detailed summary of recent advances in sample preparation methods, common detection techniques and parameters of method calibration and validation.

## 2. Pretreatment and Derivatization of Biological Samples for SCFAs

Sample extraction and purification from biological matrices is crucial for analytical implementation in terms of selectivity, sensitivity, and accuracy. The purpose of sample pretreatment is to extract and concentrate analytes of concern while eliminating interfering matrix elements, and to alleviate damage to the column caused by impurities [42]. Protein precipitation (PP) [1], solid-phase extraction (SPE) [43], and liquid-liquid extraction (LLE) [44] are conventional techniques for endogenous compound extraction to attain effective analyte recovery by lowering ion suppression and enrichment of analyte signal [42]. A key component of pretreatment is deproteinization, and can be achieved via precipitation with organic solvents, acidification, ultrafiltration and centrifugation [36,45,46].

Optimal conditions for long term stability and extraction recovery of SCFAs varies depending on the biomatrix; Samples should be stored between −20 and −80 °C with limited freeze-thaw cycles. Feces are generally analyzed immediately after excretion and can be stored in isopropanol (IPA) or ethanol (EtOH) for optimum preservation [47,48]. One study found with no change in SCFA concentration in human feces stored in 70% EtOH at room temperature for up to seven days [49]. 

Historically, LC-MS analysis of underivatized SCFAs required harsh chromatographic conditions due to their high polarity, low molecular weight, poor retention on reversed phase columns and poor ionization [37,50]. To overcome this, SCFAs may be subject to chemical derivatization, usually through coupling of the carboxylic acid with a reactive nucleophile in presence of a coupling agent and base catalyst (i.e., EDC, pyridine) at low pH. The resulting SCFA derivatives are less polar with more favorable chromatographic and mass spectral properties. There are several commercially available reagents for this purpose: 3-nitrophenylhydrazine (3-NPH), 2-picolylamine (PA), and O-benzylhydroxylamine (O-BHA), are hydrazine, amine, and hydroxylamine derivatives which share hydrophobic properties such as phenyl or benzyl groups [17,33,51]. Alkyl halides such as p-dimethylaminophenacyl (DmPA) bromide can react with carboxylic acids to form esters, but high organic solvent content is needed [52].

The use of isotopically labeled derivatization reagents can further enhance analysis of endogenous metabolites [40,52,53,54]. In a pioneering study published in 2015, Han et al. employed ^12^C/^13^C-3-NPH to convert ten SCFAs to stable 3-nitrophenylhydrazones following SPE of human fecal samples with 50% acetonitrile (ACN). Detection sensitivity in negative ESI mode UPLC-MS/MS analysis was 3- to 25-fold higher than with 2-NPH-modified SCFAs used in earlier attempts to quantify endogenous SFCAs with LC-UV [55]. The accuracy range was between 93.1% and 108.7%, with intra- and inter-day variation less than 8.8%. The average molar ratio of acetate, propionate and butyrate detected in stool samples was consistent with previous findings in portal blood [20,51]. This method has been adapted to quantify SCFAs and other metabolites in human serum, plasma and mouse biofluids and tissues, and to assess the pre-analytical stability of fecal SCFAs. Some adaptations have substituted ACN with IPA for optimal recovery and excluded pyridine to limit interference with acetic acid [1,49,56,57]. Lack of reaction quenching and adduct formation may be a limitation, though addition of formic acid has reportedly overcome this [49]. Other disadvantages of 3-NPH are limited detection of branched-chain SCFAs and inaccessibility of isotopically labeled 3-NPH [32]. Although the electronegative nitro group of 3-NPH affords compatibility with negative ESI mode MS applications, positive mode is generally preferred for its increased sensitivity. Commercially available ^12^C or ^13^C labeled aniline has been used as an alternative for derivatization of stool extracted SCFAs in ACN for 2 h at 4 °C. Reactions were quenched with succinic acid and 2-mercaptoethanol before RP LC-MS ESI positive mode analysis to eliminate derivatized residual acetic acid within the system [32]. 

Other considerations include reaction time and temperature, aqueous solubility, or if there are additional analytes of interest. Zheng et al. synthesized *N-*(4-(aminomethyl)benzyl)aniline (4-AMBA) and stable 4-AMBA-*d_5_* for labelling less abundant carbonyl, hydroxyl, and alkenyl fecal SCFAs, a 10 min reaction at 40 °C with 90% completion in presence of triphenylphosphine (TPP) and 2,2′-dipyridyl sulfide (DPDS) [58]. In a study by Zeng et al., derivatization with O-BHA outperformed three other common reagents, including 3-NPH, for simultaneous detection of ketone bodies and SCFAs from mouse feces and plasma at the sub-fmol level with positive ESI mode HPLC-MS. The reaction was carried out at room temperature, pH 5, in presence of EDC. Pyridine was excluded to avoid interference with acetic acid [44]. Jaochico et al. optimized this from a two-hour reaction time to 10 min by replacing methanolic solvents with aqueous conditions to avoid possible formation of methyl ester SCFAs. However, the use of ethyl acetate precluded quantification of acetate in the study. In evaluating the intra- and inter-day stability of plasma and urine SCFAs, they found the average change in recovery for PA, isobutryic acid (IBA), BA, 2-methylbutyric acid (2-MBA), and isovaleric acid (IVA) to be less than 15% (5 h at room temperature and three freeze-thaw cycles at −20 °C) [34]. Song et al. successfully derivatized all twelve SCFAs with 4-acetamido-7-mercapto-2,1,3-benzoxadiazole (AABD-SH) within five minutes at room temperature, whereas 3-NPH and aniline require incubation times over 30 min at temperatures below 37 °C [37]. Following esterification of deprotonated SCFAs with condensation reagent hexafluorophosphate azabenzotriazole tetramethyl uronium (HATU), Fu et al. used 4-aminomethylquinoline (4-AMQ) dissolved in ACN with aid of *N*,*N*-Diisopropylethylamine (DIPEA) to derivatize EtOH-extracted fecal SCFAs within 20 min. at room temperature, followed by LLE with methyl tert butyl ether (MTBE) [47].

Charge reversal by labeling with inherently cationic compounds can improve ionization of polar metabolites and detection sensitivity in positive ESI mode LC-MS/MS [59]. Jiang et al. took this approach further with a novel twin derivation strategy in which endogenous and internal standards were labeled with 5-(dimethylamino)naphthalene-1-sulfonyl piperazine (Dns-PP) and (diethylamino)naphthalene-1-sulfonyl piperazine (Dens-PP), respectively—an alternative to isotopically labeled standards—while introducing naphthalene for enhanced column retention and three easily protonatable moieties. After LLE of serum with ethyl acetate, a total of 30 free fatty acids, including 11 SCFAs, were quantified, with lower limit of quantifications (LLOQ) ranging from 4 nM (3-methylpentanoic acid (3-MPA), IVA) to 20 nM (PA) [60]. Wei et al. reported a rapid (3 min) derivatization for quantitation of both free and hydroxylated plasma SCFAs using deuterated N-dimethyl-6,7-dihydro-5H-pyrrolo[3,4-d]pyrimidine-2-amine (*d*_0_/*d*_6_-DHPP) and EDC in MeOH [41]. Recently, a 15 min derivatization at 20 °C was performed using novel ^12^C/^14^N and ^13^C/^15^N-labeled 5-(dimethylamino)-1-carbohydrazide-isoquinoline (DMAQ), which contains an easily ionizable tertiary amine and isotopic dimethyl amine groups for aid of separation of linear and branched SCFAs [40].

## 3. Methods for SCFAs Quantification

Practical and high-throughput analytical methods for SCFAs and their homologues are necessitated in part by increased relevance to nutritional and clinical research towards understanding the role of SCFAs in physiological processes and pathologies. Although LC-MS-based methods are the focus of this review, it should be noted that several detection techniques have been applied to quantitate biologically derived SCFAs during chromatographic separation. The chromatographic and MS parameters adapted for SCFAs and their applications are presented in Table 1, Table 2 and Table 3.

### 3.1. Chromatographic Analysis

The derivatized SCFAs demonstrated improved chromatographic separation and increased detection sensitivity. Reverse-phase HPLC (RP-HPLC) is most commonly used to identify, separate, and quantitate analytes of interests. The frequently stated stationary phase chemistry is C18 [34,37,39,49], although dimethylphenyl-hexyl columns have been used for enhanced retention of polar aromatic compounds [61]. Normal phase, or hydrophilic interaction LC (HILIC) columns may be called for, such as with Girard’s reagent T (GT)-labeled SCFAs which lack the degree of hydrophobicity rendered by derivatizing agents with cyclic functional groups [38], or to maintain peak shape after injection of high organic content samples [62]. Mobile phases generally consist of acidified HPLC-grade polar (water) and non-polar solvents (i.e., ACN, MeOH) for analyte separation by isocratic or gradient elution profiles to reduce probable interference in complex bio-matrices. An acidic mobile phase pH can increase peak height and optimize peak shape and can be achieved by addition of formic acid. Derivatizing agent, column pore size, and flow rate will influence chromatographic separation time; For example, baseline separation of all NPH-SCFAs could be achieved in 13 min, and BHA-derived SCFAs in six minutes with flow rate 0.4 mL/min on C18 columns with pore sizes 1.7 μm and 2.7 μm, respectively [44,51]. Elution profile should be optimized based on application; a steeper gradient will accelerate run time but may compromise baseline separation. Song et al. reported a 65 min baseline separation of all 12 AABD-SH labeled-SCFAs. After reducing analysis time to 35 min with a steeper gradient, four branched SCFAs could not be quantified [37].

### 3.2. Detection Methods

Literature from last two decades showed that gas chromatography (GC) or liquid chromatography (LC) coupled to mass spectrometric detector (MS or MS/MS), ultraviolet detection or flame ionization detection [43] with ESI have been majorly employed to detect SCFAs in biological samples such as stool [32,37,49,51], plasma [34,37], blood [61], and cell culture media [38]. High-performance liquid chromatography with diode-array detector (HPLC-DAD) has been used for the direct quantification of SCFAs, however this method involves complex sample purification to decrease probable chromatographic interferences from endogenous compounds [63]. The MS method overcomes the requirement of complex sample preparation process, but it has its own constraints comprising relative expensive equipment and possible matrix effects interference in biological samples [64]. The utilization of LC-MS/MS based bioanalytical methods to define trace level of SCFAs in bio-fluids has significantly increased due to its distinct enhanced sensitivity, and specificity [44,51]. The use of UV detection is a low-cost method for quantitation, but it has been largely replaced by LC-MS/MS recently due to the necessity of high analyte concentration in a restricted source of biological samples [65,66]. In particular, ultra-high-pressure LC (UHPLC) shortens the typical analysis time of 45–60 min needed with GC to as short as 15 min without compromising chromatographic resolution [32].

**Table 1 ijms-23-13486-t001:** SCFAs quantified in human blood, plasma, and serum.

Method	Ionization Source	SCFAs	BioMatrix (Volume)	Analytical Column	Analysis Run Time (min)	Mobile Phase	Linearity	Extraction Solvent/Method	Derivatizing Agent & Conditions	Internal Standard	Ref.
LC-MS/MS Single reaction monitoring (SRM)	Heated-ESI	BA, HA	Blood (25–100 μL)	Ascentis Express Phenyl-Hexyl (5 cm × 2.1 mm, 2.7 μm)	10	A: 5 mM ammonium acetate; B: IPA; C: MeOH	0.12–10,000 pmol	Saponification KOH-EtOH 80 °C, 45 min	2-NPH/EDC 20 min, 60 °C	undecanoic acid	[61]
LC-MS/MS Single ion monitoring (SIM)	ESI negative mode	BA, VA, HA and branched forms	Serum (50 μL)	Restek Raptor C18 (2.1 × 100 mm, 2.7 μm)	18	A: 0.1% FA in water; B: ACN	10–2500 ng/mL	IPA	3-NPH Pyridine/EDC in MeOH 37 °C, 30 min	2-isobutoxyacetic acid	[1]
LC-MS/MS Multiple reaction monitoring (MRM)	ESI positive mode	PA, IBA, BA, 2-MBA, IVA, VA, 3-MVA, IHA, HA	Plasma	Kinetex Evo C18 (50 × 2.1 mm, 1.7 μm)	7.5	A: 0.1% FA in water; B: 0.1% FA in EtOH		Ethyl acetate	O-BHA EDC/pyridine RT, 10 min	d_3_-AA, d_3_-PA, d_2_-BA, d_3_-IBA, d_2_-VA, d_2_-IVA, d_2_-HA	[34]
LC-MS/MS	ESI positive mode	12 SCFAs	Exhaled breath condensates (1 mL)	Pursuit 5 C18 (150 × 2.0 mm)	35	A: 0.1% FA in water; B: 0.1% FA in ACN	0.1–1000 μM	Water	AABD-SH TPP/DPDS in DCM RT, 5 min	d_3_-AA, d_6_-AA, d_7_-BA, d_4_-VA, d_5_-HA	[37]
LC-MS/MS	ESI negative mode	AA, PA, BA	Plasma (50 μL)	Alltima ODS RP (250 × 2.1 mm, 5 μm)	25	A: 1.5 mM HCl; B: 5% EtOH in water (0.75 mM HCl)	0–250 μM	100% MeOH		75 μM AA, BA, PA	[50]
HPLC-MS/MS	ESI positive mode	Free and esterified AA, PA, BA, IBA, 2-MBA, VA, IVA, HA	Plasma (50 μL)	Acquity UPLC HSS T3 (2.1 × 100 mm, 1.8 μm)	15	A: 0.1% FA in water; B: 0.1% FA in IPA	1–10,000 nmol	HClO_4_/NaOH 90 °C, 20 h	Glycine ethyl ester/EDC 2 h	^13^C-PA and ^13^C-BA; deuterated AA and VA	[67]
LC-MS/MS (MRM)	ESI negative mode	AA, PA, IBA, BA, IVA, VA, IHA, HA	Serum	ACE C18-AR (100 × 2.1 mm, 1.7 μm)	15	A: water B: ACN	0.015–25 μg/mL	ACN/methyl tert-butyl ether (MTBE)	3-NPH-HCl/EDC-HCl 40 °C, 30 min	^13^C_2_-AA, d_7_-BA, 2-EtBA	[68]
UHPLC-MS/MS	ESI negative mode	AA, PA, BA, IBA, 2-MBA, IVA, VA	Plasma	ZORBAX Eclipse Plus C18 (2.1 × 100 mm, 1.8 μm)	20	A: 0.1% FA in water; B: 0.1% FA in IPA:ACN (3:1, *v*/*v*)	5–2000 ng/mL (300–30,000 ng/mL for AA)	MeOH	3-NPH/EDC 40 °C, 20 min	isotope-labeled derivatized with ^13^C_6_-3-NPH	[57]

**Table 2 ijms-23-13486-t002:** SCFAs quantified in human feces and urine.

Method	Ionization Source	SCFAs	BioMatrix (Volume)	Analytical Column	Analysis Run Time (min)	Mobile Phase	Linearity	Extraction Solvent/Method	Derivatizing Agent & Conditions	Internal Standard	Ref.
LC-MS/MS (MRM)	ESI positive mode	PA, IBA, BA, 2-MBA, IVA, VA, 3-MVA, IHA, HA	Urine	Kinetex Evo C18 (50 × 2.1 mm, 1.7 μm)	7.5	A: 0.1% FA in water; B: 0.1% FA in EtOH		Ethyl acetate	O-BHA EDC/pyridine RT, 10 min	d_3_-AA, d_3_-PA, d_2_-BA, d_3_-IBA, d_2_-VA, d_2_-IVA, d_2_-HA	[34]
LC-MS/MS (MRM)	ESI positive mode	12 SCFAs	Feces (250 mg)	Acquity UPLC HSS T3 (2.1 × 100 mm, 1.8 μm)	15	A: 0.1% FA in water; B: 0.1% FA in IPA	1–10,000 nmol	50% ACN	^12^C-/^13^C-aniline EDC 4 °C, 2 h (quenching: succinic acid/2-mercaptoethanol; 4 °C, 120 min)	d_5_-benzoic acid	[32]
LC-MS/MS (MRM)	ESI negative mode	AA, PA, BA, IBA	Feces (2 mg DW/mL)	Kinetex XB-C18 (50 × 2.1 mm, 2.6 μm)	4	A: 0.1% FA in water; B: 0.1% FA in ACN		70% IPA	3-NPH/EDC FA 40 °C, 30 min	[^13^C,d_3_]-AA, d_5_-PA, d_7_-BA	[49]
UPLC-MS/MS (MRM)	ESI negative mode	10 SCFAs	Feces (2 g)	Waters BEH C18 (2.1 × 100 mm, 1.7 μm)	11	A: 0.1% FA in water; B: 0.1% FA in ACN	0.12–2500 nM	50% ACN	^12^C/^13^C_6_-3-NPH EDC/pyridine 40 °C, 30 min	Stable isotope-labeled versions derivatized with ^13^C_6_-3-NPH	[51]
LC-MS	ESI positive/negative switching	AA, PA, BA	Urine (100 μL)	ZICHILIC (4.6 × 150 mm, 3.5 μm)	25	A: 0.1% FA in water; B: 0.1% FA in ACN	0.05–1.6 μg/mL	THF:water (1:1)	N,N-dimethyl-p-phenylenediamine (DPD)/EDC 60 °C, 45 min	^13^C_2_-AA, ^2^H_2_-PA, ^2^H_5_-BA	[62]
UPLC-MS/MS (MRM)	ESI positive mode		Feces (3 g)	Waters BEH C18 (2.1 × 100 mm, 1.7 μm)	5.5	A: 0.1% FA in water; B: 0.1% FA in MeOH	1.5–10,000 μM	EtOH	HATU 4-AMQ/DIPEA (1000:1, *v*/*v*) in ACN RT, 20 min LLE: Dulbecco’s phosphate buffered saline/MTBE	d_4_-AA (surrogate analyte), stable isotope labeled standards	[47]
LC-MS		AA, BA	Urine		~15 min			ACN/ Triethanolamine	^12^C/^13^C-DmPA 90 °C, 60 min (Quenching: TPA)	113 carboxylic acid standard library	[52]

**Table 3 ijms-23-13486-t003:** SCFAs quantified in mouse/non-human blood, plasma, and feces.

Method	Ionization Source	SCFAs	BioMatrix	Analytical Column	Analysis Run Time (min)	Mobile Phase	Linearity	Extraction Solvent	Derivatizing Agent & Conditions	Internal Stds	Ref.
LC-MS/MS	ESI positive mode	12 SCFAs	Feces (20–25 mg), plasma (200 μL)	Pursuit 5 C18 (150 × 2.0 mm)	35	A: 0.1% FA in water; B: 0.1% FA in ACN	0.1–1000 (μM)	Water	AABD-SH/ TPP/DPDS in DCM RT, 5 min	d_3_-AA, d_6_-PA, d_7_-BA, d_4_-VA, d_5_-HA	[37]
LC-MS/MS (MRM)	ESI positive mode	10 SCFAs	Feces (100 mg), plasma	Kinetex C18 (100 × 2.1 mm, 2.6 μm)	5.3	A: 0.1% FA in 10 mM NH_4_COOH; B: 0.1% FA in MeOH:IPA (9:1 *v*/*v*)	0.04–250 μM	50% MeOH; DCM	O-BHA/EDC 25 °C, 1 hr	d_5_-PA, d_7_-BA, d_1_-HA	[44]
UHPLC-MS/MS	ESI positive mode	AA, PA, BA, IBA, VA, IVA, HA, IHA, SA	Feces (2 mg)	Acquity BEH C18 (2.1 × 100 mm, 1.7 μm)	10	A: 0.1% FA in water; B: 0.1% FA in MeOH		MeOH QuEChERs method	2-picolylamine/ DPDS/TPP 60 °C, 10 min	d_4_-AA, d_6_-PA, d_9_-VA, d_11_-HA	[69]
UHPLC-MS/MS	ESI positive mode	21 SCFAs (alkyl, carbonyl, hydroxy, and alkenyl)	Feces (20 mg)	Acquity UPLC BEH C18 (50 × 2.1 mm, 1.7 μm)	45	A: 0.1% FA in water; B: ACN	0.015–1000 ng/mL	Liquid nitrogen evaporation; HCl pH 1.0 Ether	4-AMBA and 4-AMBA-d5 40 °C, 10 min	4-AMBA-d_5_ labeled SCFAs	[58]
UHPLC-LTQ-Orbitrap MS	ESI negative mode	AA, PA, BA, IBA, VA, IVA, HA, 2-MBA	Bronco- alveolar lavage fluid (2.4–3 mL)	Acquity BEH C18 (2.1 × 100 mm, 1.7 μm)	13.5	A: 0.01% FA in water; B: 0.01% FA in ACN	0.05–15,360 nmol/L	ACN	3-NPH EDC/pyridine 40 °C, 30 min	d_4_-AA	[56]
LC-MS/MS (MRM)	ESI positive mode	AA, PA, BA, VA, IVA, HA, IHA	Hamster plasma (50 μL) & feces (0.2 g)	xBridge C18 (100 × 2.1 mm, 3.5 μm)	27	A: 0.1% FA in water; B: MeOH	1–50,000 ng/mL	ACN	2-bromo-acetophenone (BP) 40 °C, 20 min	Methyl 2-methyl-1-oxo-1,2,3,4-tetrahydronaphthalene-2-carboxylate	[39]
LC-MS (full scan/SIM)	ESI positive mode	2,2-dimethylbutyrate (DMB)	Rat plasma (200 μL)	Synergi Hydro-RP (2 × 100 mm, 4 μm)	30	A: 0.1% FA in ACN; B: 0.1% FA in water	100–10,000 ng/mL	CAN	benzylamine; deoxo-Fluor/ ^i^Pr_2_NEt, DCM −20 °C, 20 min	Dimethylvaleric acid (DMV)	[70]
UHPLC-MS/MS Parallel reaction monitoring (PRM)		10 SCFAs and 30 OH-SCFAs	Feces (20 mg), serum (50 μL), liver tissue (5 mg)	Phenomenex polar C18 column (1.6 μm, 2.1 × 150 mm)	14	A: 0.1% FA in water; B: ACN	0.1–200 ng/mL	MeOH	d_0_/d_6_-DHPP EDCI <30 °C, 10 s	Derivatized with d_6_-DHPP	[41]
LC-MS	ESI positive mode	AA, PA, BA, VA, IBA, 2-MBA, IVA	Piglet plasma (100 μL), cecum digesta (100 mg)	ACQUITY UPLC BEH C18 column (2.1 × 150 mm, 1.7 μm)	7	A: 0.1% FA in water; B: 0.1% FA in ACN	0.1–500 μg/L	80% MeOH	^12^C/^14^N or ^13^C/^15^N-DMAQ EDC/HOBt 20 °C, 15 min	Stable isotope-labeled	[40]
LC-MS/MS (SIM)	ESI positive mode	AA, PA, BA, IBA, VA, IVA, 2-MBA, IHA, 3-MPA, HA, 2-MHA	Rat serum	XDB-C18 (2.1 × 100 mm, 1.8 μm)	~20	A: 0.1% FA in water; B: MeOH	0.02–100 μM	ethyl acetate/FA	Dns-PP/HATU 37 °C, 150 min	Dens-PP labeled standards	[60]

## 4. Calibration Approaches for Quantification of SCFAs

Whilst developing chromatographic methods, the availability of a suitable blank matrix for quantitation of endogenous compounds always is a main concern. Different methodologies are applied by scientists in general to combat the complications related to blank matrices involving, standard addition [54], background subtraction [54], and by utilizing stripped matrices [71,72,73], artificial matrices [74] and surrogate analytes [71]. In the standard addition method, the native matrix is spiked with varying concentrations of standards for the creation of calibration curve [67]. This allows quantitation of exogenous analytes for each individual sample, although time consuming and requires larger sample volumes. With background subtraction, endogenous analytes are subtracted from known standards in a pooled native matrix prior to calibration curve construction. This may be limited by irreproducibility due to batch and sample variability. Surrogate matrices, either artificial or stripped, mimic properties of the authentic matrix (i.e., extraction recovery, matrix effect) while lacking target analytes. Saline solutions are common artificial matrices, whereas a stripped matrix can be prepared by stripping the native biomatrix of endogenous components using UV treatment or activated charcoal [34,60]. Alternatively, a blank matrix, consisting of mobile-phase solvent may be used. Stable-isotope-labeled standards as surrogate analytes can also be applied for direct and sensitive quantification of analytes [34]. These have minimal variations in terms of signal intensity, extraction recovery, and chromatographic retention as endogenous analytes of interest. The use of mentioned calibration approaches for SCFA quantitation has been reported extensively and reviewed in more detail elsewhere [1,32,33,34,37,41,60,69,75,76].

## 5. SCFA Quantitation in Different Biological Matrices

### 5.1. Human Blood, Plasma and Serum

The concentration of SCFAs in the large intestine is approximately 1000-fold higher than in peripheral blood, yet up to 95% of SCFAs can be absorbed, either shuttled to the liver through the portal vein or directly into bloodstream [15]. Quantitative LC-MS reports of SCFAs in human blood, plasma and serum are limited and reflect a wide profile range among healthy heterogeneous populations. This variability depends on individual factors such as metabolic age, hormones, and diet [1,61]. Various pretreatment and LC-MS parameters used for detection of SCFAs in human serum and plasma can be found in Table 1.

Sampling 547 human subjects, Chen et al. quantified free and esterified forms of BA and HA with LC-MS following derivatization by 2-NPH. With LLOQs of 5 pmol and 0.1 pmol, concentrations of free BA and HA were consistent with earlier LC- and GC-MS analyses, showing no statistical difference between age or sex [61]. In a 12-week long study conducted by Sowah et al., weight-loss associated changes in plasma SCFAs (free and esterified) were monitored in 150 overweight and obese adults. They found a weak inverse correlation between nutrient intake and AA, BA, and VA levels, but AA to be significantly higher than previously reported in adults. This was attributed to either an artifact of derivatization with glycine ethyl ester and/or multiple freeze-thaw cycles leading to acetyl ester hydrolysis [67]. These analyses of esterified fatty acids required saponification, a hydrolysis reaction facilitated by a strong base at high temperatures.

Dei Cas et al. performed the IPA protein precipitation coupled LC-MS/MS analysis of short and medium fatty acids in human serum samples utilizing chemical derivatization using 3-NPH to the respective acylhydrazones. Serum concentrations ranged from 62.9 ± 15.3 (VA), 1155.0 ± 490.4 (IVA), 935.6 ± 246.5 (BA), 698.8 ± 204.7 (IBA), and 468 ± 377.5 (HA) ng/mL respectively (mean ± SD) [1]. Using this approach, Liao et al. assessed plasma SCFAs, bile acids, and tryptophan metabolites in patients with cardiovascular disease and found decreased levels of circulating BA compared to age/sex matched controls [57]. SCFAs in plasma of five patients with inflammatory bowel disease (IBD) were analyzed post O-BHA derivatization with Jaochico et al.’s method, which included stable isotope-labeled internal standards (with the exception of 2-MBA, 3-MVA, and IHA, for which d_2_-IVA and d_2_-HA were used), surrogate matrices for calibration, and MRM transitions in positive ESI mode. A wide distribution of plasma SCFAs among individuals were reflected in the results, with higher average levels of all but IVA in the IBD group compared to controls. However, 3-MVA levels were extrapolated separately due to an unknown interfering peak [34].

### 5.2. Human Urine and Feces

Rapid quantification of SCFAs and metabolites in urine and feces can accelerate our understanding of the metabolism and clearance process in vivo. Urine and feces sampling poses several advantages such as easy collection procedure, sufficient specimen volume and non-invasive procedure, but analysis is difficult due to a variety of interfering compounds. Table 2 summarizes the published LC-MS analyses of SCFAs in human urine and feces. One of the first reports of LC-MS detection of urine SCFAs came out in 2017. Urine of patients with active or remissive ulcerative colitis (UC) was labeled with isotope-coded *N*,*N*-dimethyl-*p*-phenylenediamine hydrochloride (DPD). They found higher BA levels in UC urine compared to control samples, but noted a high abundance of AA as a potential result of environmental SCFA contamination; Only AA, PA, and BA were assessed [62]. Subsequently, Jaochico et al. applied their robust multi-matrix method for quantitation of seven O-BHA-derived SCFAs from urine of IBD patients and healthy volunteers. Compared to charcoal stripped plasma, the calibration slope measured in stripped urine deviated less from the surrogate matrix (water) slope, although both fell within the acceptable criteria (less than 15%) [34].

Han et al. reported significantly increased fecal branched SCFAs in a patient with Type II diabetes compared to five healthy controls after UHPLC-MS/MS analysis of ^12^C/^13^C-3-NPH-derived SCFAs [51]. Using the same pretreatment method, another group identified a positive correlation between branched SCFA concentrations in infant feces and breastmilk lipid and human milk oligosaccharide (HMO) content [77]. Liebisch et al. examined the pre-analytic stability of PA, AA, BA, and IBA in human feces following 3-NPH derivatization. They found that SCFA concentrations may change substantially within hours, implicating differences between the individual donors. Reproducibility (%CV less than 10%) was achieved by the application of stable isotope labelled internal standards [49]. Chan et al. detected and quantified a total of nine SCFAs and characterized the relative abundance of AA, PA, and BA in human infant feces during the first 12 months via LC-MS/MS following derivatization of endogenous and spiked SCFAs with ^12^C- and ^13^C-aniline. Note, the authors did not measure extraction efficiency, but adjusted volume of extraction solvent based on stool weight [32].

### 5.3. Mouse Plasma, Serum, and Feces

There are several published reports of SCFA detection via LC-MS-based methods in non-human biomatrices. Animal model systems are commonly used for preclinical PKPD method validation [78]. Table 3 summarizes the representative examples for SCFAs quantification in non-human blood, plasma, and feces published within the last 15 years.

In an early study Parise et al. quantified 2,2-dimethylbutyrate (DMB) in rat plasma following derivatization to benzylamide [70]. DMB and other branched SCFAs are promising drug candidates for hemoglobinopathies and have been shown to stimulate fetal globin and erythropoiesis in multiple species [79,80,81,82,83]. Their method was found suitable for assessing the PK of DMB in supplement to toxicology studies.

Animal models have been utilized in several studies of diet- or disease-induced metabolic changes. Nagatomo et al. developed a UHPLC-MS/MS method for the simultaneous determination of SCFAs in feces of diabetic obese mice following extraction and coupling with 2-PA. The derivatized analytes showed limit of detection (LOD) less than 75 nM, limit of quantitation (LOQ) less than 100 nM. Preliminary analyses revealed decreased branched chain SCFAs in diabetic mice, while high-fat diet treatment increased succinic acid levels for both control and diabetic mice over a nine-month period [69]. Song et al. observed an increase in BA and VA in mice fed with high-fat diets compared to low-fat following derivatization of mouse plasma and feces with AABD-SH [37]. Ma et al. found drastically increased BA levels in stool samples of hyperlipidemia hamsters after treatment with berberine and metabolites; their use of derivatizing agent 2-bromoacetophenone prior to LC-MS/MS was 200 to 2000-fold more sensitive than GC-based methods, and resulted in LLOQ of BA less than 15 nM [39]. Liebisch et al. examined fecal SCFA levels in mice fed a non-fermentable diet compared to normal and found a significant decrease, as expected [49]. Zeng and Cao validated their LC-MS/MS method for SCFA and ketone body detection in lean and diet-induced obese (DIO) mice plasma and feces, finding acetate levels significantly increased after periods of fasting [44]. Moreover, 3-NPH derivatization has been applied to compare fecal SCFAs of specific pathogen free (SPF) mice and antibiotic-fed mice [84], and for detection of SCFAs in bronchial alveolar lavage fluid, serum, feces and lung tissue of asthmatic mice using ultra-sensitive UHPLC-LTQ-Orbitrap MS [56]. The novel twin derivatization strategy by Jiang et al. was applied to quantify 11 serum extracted SCFAs in a rat nephrotoxicity and gastrotoxicity model [60].

Using UHPLC-MS/MS, Wei et al. measured free and hydroxylated SCFAs in mouse serum, feces, and liver tissue from breast cancer nude mice to monitor mutagen-induced metabolic changes. Their application of high-resolution PRM mode provided a feasible method to locate unidentified SCFAs species and limit false positives [41]. Zheng et al.’s novel 4-AMBA/4-AMBA-d_5_ labeling procedure enabled UPLC-ESI-MS/MS quantification of 21 fecal SCFAs from Alzheimer disease (AD) mice. The LODs of SCFAs were as low as 0.005 ng/mL and good linearity of 34 SCFAs were obtained (R^2^ = 0.9846–0.9999), indicating the acceptable reproducibility of this highly sensitive method. The study revealed significant differences in levels of nine SCFAs between healthy and AD mice (n = 28 for each group) [58].

### 5.4. Bacterial Cell Culture

In addition to biofluids and tissues derived from humans and model organisms, it is arguably as important to measure prokaryotic production of SCFAs. Kim et al. developed a GC-FID method for microbial samples in which SCFAs were extracted with diethyl ether-hexane and aminopropyl SPE and resolved on an acid-modified poly(ethylene glycol) capillary GC column. Their study included AA, PA, VA and BA, with pivalic acid as an internal standard [43]. The average recoveries ranged from roughly 96.5 to 109% (RSD less than 4.08%). While this technique enables use of the most common GC detector without derivatization, note the use of diethyl ether may result in overestimation of acetic acid due to hydrolysis [85]. In 2020, Song et al. demonstrated a HILIC-based LC-MS/MS derivatization method to quantify SCFAs produced by *Eubacterium rectale* (*E. rectale*), which makes up 13% of gut microbiota, from a small amount of total extracellular metabolites [41]. AA, PA, BA, VA and HA were quantified in 5 to 20 μL aliquots of culture medium. The production of BA was also successfully monitored in a co-culture of *E. rectale* and *Biofidobacterium longum* by GT-labeled BA [38]. Recently, Bihan et al. adapted the use of ^13^C/^12^C-labeled aniline for RP LC-MS quantitation of SCFAs produced by five microbial species. They presented an isotope dilution strategy to account for potential derivatization-based error. Inter- and intra-day precision of the method was reported to be less than 1% and 2–3%, respectively [86]. These methods provide a basis for future studies involving gut microbial fermentation of SCFAs.

## 6. Conclusions

A comprehensive knowledge of the functional roles of SCFAs in humans is needed to demystify the relationship between the gut microbiome, nutrition, and pathophysiological conditions. An assortment of chromatography- and spectrometry-based methods for determination of SCFAs in biomatrices have been developed within the last two decades. The incorporation of suitable extraction and derivatization techniques for LC-MS applications has significantly enhanced the acquisition of quantitative data for SCFAs, in both human and nonhuman systems. Ultimately, a reliable, reproducible, and cost-effective analytical method for SCFAs and other metabolites is key for diagnosis and preventative detection in a clinical setting.

## Data Availability

Not applicable.

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
