# Peer review of "A Concise Review of Liquid Chromatography-Mass Spectrometry-Based Quantification Methods for Short Chain Fatty Acids as Endogenous Biomarkers"

_ijms, 2022, doi:10.3390/ijms232113486_

Round 1

Reviewer 1 Report

This concise review paper presents the current analitical progress of SCFAs in different biological matrices mainly by using derivitized reagents. sample pretreatment processes and choromatographic separation were also included delibrately. Furthermore, authors dicussed the pros and cons of different detection methods of SCFAs such as GC, LC-UV and LC-MS and the concerning problems facing the field like the blank matrix availability in detail.

Yet there are still some minor problems should be revised as follow:

1. the body of the paper should be restructured such as the numering of the main points(as shown in the paper: 1.inroduction; 1. Pretreatment and derivatization of biological samples for SCFAs;1. Methods for short chain fatty acids quantification. etc. ).

2. Extraction solvents of CAN in Table 3 should be corrected as acetonitrile (ACN)

Author Response

Reviewers’ Comments to the Authors:

Reviewer 1

“This concise review paper presents the current analitical progress of SCFAs in different biological matrices mainly by using derivitized reagents. sample pretreatment processes and choromatographic separation were also included delibrately. Furthermore, authors dicussed the pros and cons of different detection methods of SCFAs such as GC, LC-UV and LC-MS and the concerning problems facing the field like the blank matrix availability in detail

Yet there are still some minor problems should be revised as follow:

  1. the body of the paper should be restructured such as the numering of the main points(as shown in the paper: 1.inroduction; 1. Pretreatment and derivatization of biological samples for SCFAs;1. Methods for short chain fatty acids quantification. etc. ).
  2. Extraction solvents of CAN in Table 3 should be corrected as acetonitrile (ACN)”

Authors’ response:

Thank you for noticing these errors.

1) The incorrect numbering for each section and subsection within the manuscript has been corrected (1, 2, 3, 3.1, 3.2, etc.).

2) Extraction solvents listed as “CAN” in Table 3 have been corrected to “ACN”

In response to your suggestion of extensive editing of the English style and language: we have carefully looked through the manuscript and attempted to improve the fluency of the text.

Reviewer 2 Report

Summary

The authors describe a substantially updated overview of sample pretreatment techniques and LC-MS-based methods for quantitative analysis of SCFAs in blood, plasma, serum, urine, feces and bacterial cultures. They present a detailed summary of recent advances in sample preparation methods, common detection techniques and parameters of method calibration and validation.

In my opinion, the review is clear and comprehensive enough, and of relevance to the field. There are no similar reviews published recently.

Major Issues

There is ambiguity about the nomenclature of SCFAs (see comments in the manuscript itself, in the "Abbreviations" section). I think it is very important that this aspect be remedied.

Minor Issues

They are attached in the manuscript itself.

Author Response

Summary: The authors describe a substantially updated overview of sample pretreatment techniques and LC-MS-based methods for quantitative analysis of SCFAs in blood, plasma, serum, urine, feces and bacterial cultures. They present a detailed summary of recent advances in sample preparation methods, common detection techniques and parameters of method calibration and validation.

In my opinion, the review is clear and comprehensive enough, and of relevance to the field. There are no similar reviews published recently.

Major Issues: There is ambiguity about the nomenclature of SCFAs (see comments in the manuscript itself, in the "Abbreviations" section). I think it is very important that this aspect be remedied.

Minor Issues: They are attached in the manuscript itself

Authors’ response:

Regarding the nomenclature of SCFAs: We completely agree with this assessment and have revised such incongruities as necessary (example: pg. 18 in word doc, changed “acetate, propionate, and butyrate” to “AA, PA, and BA”), except for the introduction. Instances of caproic acid (CA)/derivatives have been replaced with hexanoic acid (HA), as they are the same acid. Instances of “3-MBA” or “4-MVA” have been replaced with “IVA” or “IHA.” The Abbreviations section has been updated to remove any redundancies or abbreviations not present in the text.

Addressing the highlighted comments in the pdf version:

The title has been revised to “A Concise Review of Liquid Chromatography-Mass Spectrometry-Based Quantification of Short Chain Fatty Acids as Endogenous Biomarkers.” We included the full definition of LC-MS and its abbreviated form in the abstract.

We replaced any instances of hyphenated “short-chain fatty acids” with “short chain fatty acids”

Any instances of “et al” have been corrected to “et al.”

A space has been added between any temperature and degree Celsius.

In Tables 1-3, any solvents or MS methods listed without previously being defined have been defined, either in the table or previously in the text (example: 4-AMQ, HATU, DIPEA have all been defined on page 7 in word doc.). Abbreviations for reagents and solvents have been added to the abbreviations section.

On page 5 in the word document, the section of text beginning “Optimal conditions for long term stability..” has been moved as a separate paragraph.

In section 5.4 (Bacterial cell culture), we included the method developed for GC-FID mainly due to limited available studies. An additional LC-MS based study has since been included in this section.